# Universities without Walls: A Blended Delivery Approach to Training the Next Generation of HIV Researchers in Canada

**DOI:** 10.3390/ijerph17124265

**Published:** 2020-06-15

**Authors:** Francisco Ibáñez-Carrasco, Catherine Worthington, Sean Rourke, Colin Hastings

**Affiliations:** 1MAP Centre for Urban Health Solutions, St Michael’s Hospital, Toronto, ON M5B 1W8, Canada; 2Public Health and Social Policy, University of Victoria, Victoria, BC V8W 2Y2, Canada; 3Li Ka Shing Knowledge Institute of St. Michael’s, Toronto, ON M5B 1W8 Canada; sean.rourke@utoronto.ca; 4Department of Sociology, York University, Toronto, ON M3J 1P3, Canada; chastings21@gmail.com

**Keywords:** graduate training, online education, eLearning, HIV, distance education, adult education, research training, CBPR, CBR

## Abstract

(1) Background: Although HIV has not diminished in importance in Canada, the field of HIV research remains small, and the graduate students who decide to pursue careers within it feel isolated and uncertain about their professional skills and opportunities. Universities Without Walls (UWW) was created in 2009 to help redress these shortcomings. This paper presents a case study of UWW, a non-credit training program for emerging HIV researchers in Canada. In particular, we focus on the possibilities of experiential learning via online and blended delivery. UWW uses both online and in-person teaching modalities to teach engaged scholarship, interdisciplinarity, community-based research (CBR), intervention research, and ethics. (2) Methods: Using a case study, we elucidated the research question: “What are the factors that make Universities Without Walls a viable training environment in the contemporary HIV/AIDS field?” Focus groups were conducted with 13 UWW key stakeholders in 2012 during a program mid-point evaluation; in 2014, telephone or in-person interviews with the three directors were conducted by a UWW fellow (the 4th author of this paper), and in 2019 the authors analyzed the information and anecdotal evidence, which had been incorporated as thick description. In addition, fellows’ self-assessments via portfolio and results from formal learning assessments were included. We also thematically analyzed 65 student self-reports (2009–2015). (3) Results and Discussion: Each UWW cohort lasted 9 months to one year and was comprised of: a) sustained mentorship from the co-directors (e.g., phone conversations, assistance with grant writing, letters of reference, etc.); b) fortnightly online webinars that aim to develop fellows’ knowledge of community-based research (CBR), research ethics, intervention research, and interdisciplinary research; c) community service learning in the form of a “field mentoring placement”; d) face-to-face engagement with fellows and mentors, most notably at the week-long culminating learning institute; e) a stipend for fellows to carry out their training activities. The UWW pedagogical framework features experiential learning, critical pedagogy, and heutagogy made manifest in the field mentoring placements (community service learning), mentorship mediated by technologies, and in-person learning institutes. Our analysis showed that experiential learning was imparted by UWW’s a) transparency about its “implicit curriculum”, the attitudes, values, character, and professional identity imparted in the program as well as the overarching programmatic elements, such as commitment to diversity, the inclusion of those with lived experience, the flexible admissions policies and procedures, interdisciplinary faculty, flexible team, administrative structure, and valuing of technology in conducting research, learning, and teaching; b) curriculum co-designing and co-teaching, and c) sustaining a community of practice. The main results reported in our case study included significant “soft outcomes” for UWW fellows, such as developing a “social presence” as a precursor to lasting professional connections; learning to experience community-based research, intersectionality, and interdisciplinarity by interacting online with persons living with HIV, leaders in the field, and a variety of stakeholders (including nonprofit staff and policymakers). (4) Limitations: While fellows’ self-evaluation data were collected by an independent assessor and anonymized to the extent this was possible, the co-authors inevitably bring their preconceptions and positive biases to UWW’s assessment. As UWW was developed to function outside of traditional academic structures, it is unlikely that the UWW program could be transferred to a post-secondary environment in its entirety. UWW was also built for the socio-political environment of HIV health research. (5) Conclusions: The experiences of those involved with UWW demonstrate that explicit curricular components—such as interdisciplinarity, community-based research, intervention research, and applied ethics—can be learned through a blended delivery when combined with opportunities to apply the knowledge in ways, such as a field mentoring placement and a learning institute. Related to this outcome, our case study describes that implicit curricular components in the formation of a professional—the sense of self in the field as a researcher, student, and community member—can also be delivered through a blended model. However, the tools and activities need to be tailored to each student for their context, while pushing their disciplinarian and professional boundaries.

## 1. Introduction

This paper presents a case study of an innovative training program for emerging HIV researchers in Canada: Universities Without Walls (UWW, www.uwow.ca). In particular, we focus on the possibilities of experiential learning via online and blended delivery. The UWW training program is the home for many HIV research-related training activities and outputs, which are showcased on its website (uwow.ca). The UWW fellowship is a non-credit, research network affiliated training program that offers modest, project-based training funding; experiential learning activities; paired fellow-faculty mentorship; co-designed webinar and face-to-face workshop learning opportunities for graduate and post-graduate students from the humanities, social sciences, clinical sciences, and the helping professions and for HIV community leaders. The fellowship offers an intellectual and practical playground for emerging researchers to try ideas and activities without conventional fail/pass or credit pressures. UWW fellows plan their experiences, their activities, and outcomes within a nine to twelve months structure.

UWW has been delivered since 2009 and uses both online and in-person teaching modalities. Pedagogically, UWW is grounded in the notions of experiential learning and blended delivery (discussed later in this paper) as applied to a number of content areas in HIV research: engaged scholarship, interdisciplinarity, community-based research (CBR), intervention research, and ethics. In this paper, we described the application of these pedagogical concepts to the UWW training program using the case study method. The research question guiding our case study was: “What are the factors that make Universities Without Walls a viable training environment in the current HIV/AIDS field?” As well, we explored whether experiential learning in CBR is possible through online delivery. Hence, in this paper, we explain how all these elements are combined in the UWW training program.

## 2. Case Study Method

The case study method was chosen as the most appropriate approach to describe the richness and complexity of factors at play in the UWW program. Gary Thomas defined a case study as a design and analytical frame for research inquiry that incorporates a number of methods [1]. Our description of UWW and related factors gathered information from formal evaluation of UWW, our “thick description” [2,3], and our lived experience/expertise as lead educators in the program since 2009. Here, we describe the interviews and focus groups that were conducted with UWW key stakeholders during a process (mid-point) evaluation of UWW, and we also describe the UWW fellows’ self-evaluation components and processes from UWW cycles (1 through 6) (2009 through 2015). Additional data from subsequent cycles from 2016 to 2019 were integrated as supplemental evidence.

### 2.1. Focus Groups and Interviews

Focus groups were conducted with 13 UWW key stakeholders (including the directors of the program) in 2012 during a program mid-point evaluation, and, in 2014, in-depth interviews were conducted with the three directors of the program. The focus groups were conducted by the administrator of an allied HIV research program, and notes were taken during this guided conversation about the performance and direction of UWW to that point. In 2014, telephone or in-person interviews with the three directors were conducted by a UWW fellow (the 4th author of this paper) to provide further reflection on the design and delivery of the UWW program. Thus, perspectives were available from academics with substantial research and teaching experience in the area of HIV, including those who volunteered their time for UWW activities, policy-makers in the area of health and HIV (many of whom acted as mentors), frontline workers from the Canadian network of AIDS service organizations, and leaders living with HIV from the movement in Canada. Additional experience of the continuing UWW program had also been incorporated in this case study in the form of a thick description from the authors.

### 2.2. Fellow and Mentor Assessments

Indicators of success for training programs for research trainees and advanced professional trainees are often captured through program completion rates; counts of grants, conference presentations, and publications; career placement; networking success; leadership roles within professional organizations [4,5]. These were assessed for UWW fellows through annual reports from both current fellows and alumni, fellow self-assessment via portfolio, and through the follow-up email check-ins to gather updates and CVs from alumni [6,7,8]. For the fellows’ portfolios, in addition to the indicators of success, we asked fellows to self-assess in terms of their key knowledge, skills, and experience learning areas, using a learning scale based on Kolb’s learning styles [9,10], specifically a self-reported scale that ranges from “novice/advanced beginner” to “proficient/expert” used in pre- and post-test surveys. Within the self-assessments, fellows provided open-ended responses. We also asked fellows’ faculty mentors to provide feedback through a brief survey with four broad qualitative questions regarding their experience guiding a fellow. These rich textual commentaries on UWW were analyzed thematically.

### 2.3. Data Analysis

To describe the UWW rationale, structure, activities, and pedagogical framework of UWW (i.e., our “case”), we identified key themes from the mid-point focus groups with key stakeholders and augmented these with reflections from the program directors. To understand the learning experience from the perspective of fellows and mentors, we thematically analyzed 65 student self-reports and 37 faculty assessments of their experience volunteering for UWW (2009–2015). In each case, the UWW fellow/4th author conducted the first round of analysis, and a colleague review was performed by the first and second authors.

## 3. Results and Discussion

In this section, we describe our case: the UWW training program, including its rationale, structure, and activities (including blended delivery), and the aspects of pedagogy. After we present our case, we provide key themes about UWW from the interviews and focus groups, followed by thematic results from the UWW fellow self-assessments from the first five cohorts of UWW fellows (2009–2015). Additional lessons learned from the subsequent UWW training cycles (to 2019) have been incorporated. A discussion of the related literature is also integrated into our presentation of results.

### 3.1. The Case: The UWW Training Program—Its Rationale, Structure, and Activities

A sense of isolation and uncertainty among graduate students has been detected by scholars [11,12], and this sense of isolation is stronger for LGBTQ+ and other minority students [13]. UWW was created to provide a national network for HIV research support and mentorship to otherwise isolated students working in disparate disciplines and regions and to increase their chances to succeed academically and professionally in an increasingly complex and competitive professional field.

The aim of UWW was to develop a new generation of HIV researchers in socio-scientific disciplines (e.g., psychology, sociology, epidemiology, public health), including the helping professions (e.g., counseling, social work, nursing). It is important for advanced graduate training programs to balance the emphasis that is often placed on producing new trainees with knowledge and skills to work in an academic environment, with a commitment to producing trainees who are well connected to the community and more closely positioned to be successful in their chosen career paths [14]. Population and public health research are rooted in community work, which can be challenging for students to engage with from within their graduate departments. Graduate and post-graduate students tend to move to new post-secondary institutions where they do not have existing contacts with communities, politics, and organizations. Online training has been identified as support for nomadic scholars [15].

Between 2009 and 2015, UWW admitted an annual cohort of between 10 to 16 fellows, who were either graduate students from a diverse range of disciplines or community-based researchers active in the HIV field from across Canada. The number of fellows has varied since 2015, depending on available funding, and 20 more fellows have completed the program until spring 2019. UWW originally was funded (2009–2015) by a Strategic Training Initiative in Health Research (STIHR) grant from the Canadian Institutes of Health Research (CIHR) and was housed at the Ontario HIV Treatment Network (OHTN), as the training arm of the CIHR Center for REACH (Research Evidence into Action for Community Health) in HIV/AIDS, a national HIV research network. Leadership, faculty, and mentors were drawn from this Center. STIHR funds supported a manager/director, some evaluation and technological support, student fellowships, and face-to-face workshops and learning institutes.

After STIHR funding ended (2015), smaller cohorts of six to nine fellows have been funded with a combination of research training funds from the REACH network and with funding co-sponsored by Canadian researchers who have operating grants within which there is some funding support for students/trainees. We actively searched for these partnered opportunities once the original funding cycle was completed. Our fellows became key ambassadors, often acknowledging their tenure in the program and what it brought to their careers.

Each UWW cohort lasted 9 months to one year and was comprised of: a) sustained mentorship from the co-directors (e.g., phone conversations, assistance with grant writing, letters of reference, etc.); b) fortnightly online webinars that aim to develop fellows’ knowledge of community-based research (CBR), research ethics, intervention research, and interdisciplinary research; c) community service learning in the form of a “field mentoring placement”; d) face-to-face engagement with fellows and mentors, most notably at the week-long culminating learning institute; e) a stipend for fellows to carry out their training activities. From 2016 onwards, we have not implemented the “field mentoring placement”; instead, we recruited graduate students and community leaders already embedded in HIV research or health-related organizations.

#### 3.1.1. UWW Blend of Educational Delivery

Blended learning is “the organic integration of thoughtfully selected and complementary face-to-face and online approaches and technologies” [16] (p.148). There is no single agreed-upon model for blended learning because learning situations are unique. By using various elements from existing models, the UWW program can be described as a “high-impact” blend built from scratch [17], with clear spatial (face-to-face or online) attributes to reduce geographical distance and expand time (e.g., UWW eModules are available asynchronously), and merging self-initiated activities determined by each student (e.g., fellows chose group activities, such as critiquing each other’s projects and reading articles that contained elements of interest for all). In one UWW cycle, teams of fellows, with support of mentors, chose to develop peer-reviewed manuscripts in areas of particular interest [18,19]. The UWW curriculum integrated both online (e.g., webinar) and face-to-face components while maximizing benefits for both students and instructors. In the following sections, how UWW’s training model allowed these benefits to emerge are discussed in detail.

#### 3.1.2. UWW’s Pedagogical Framework

In this section, we briefly describe the theoretical elements that underpin the UWW pedagogy: experiential learning, implicit curriculum, co-designed curriculum, interdisciplinarity, and community of practice.

Experiential learning: The key concept of experiential learning through mentoring is grounded in adult learning theory, experiential learning theory, critical pedagogy, and heutagogy for lifelong learning theory [9,20,21,22]. Online learning “[a]lthough convenient, flexible, and without a doubt a more accessible learning experience, it can be a very solitary and isolated endeavor” [23] but it does not have to be. Fellows were often placed in community service learning situations or were part of community grounded research projects, and settings that were unfamiliar to fellows were deliberately chosen. In the latter case, UWW became a space for safe reflection of the politics, logistics, and technicalities of research, which often go unrecognized in research training. Besides, experiential learning requires taking risks and developing a tolerance for uncertainty [24].

The various components of the UWW fellowship (as outlined in the section above) offered multiple avenues for fellows to learn from experience. For instance, mentors were an integral component of UWW’s pedagogical framework. Intensive and time-consuming as it was, two or three mentors were always ready to have phone conversations, review drafts of academic work, and advise on applications to scholarships and grants. The mentorship was also stressed during face-to-face sessions scheduled at the learning institutes. The work of early-career HIV investigators frequently requires networks of collaborators, and UWW assisted fellows in cultivating a network of researchers that connected new investigators with other senior mentors in the field. Effective mentoring strengthens the skills and networks that any emerging generation of researchers require [25].

Implicit curriculum: based on Eisner’s ideas regarding the “implicit curriculum” originally explained in “The Educational Imagination” [26] and adapted to the online environment [27], at UWW, the faculty and fellows remained aware and critical of the attitudes, values, character, and professional identity imparted in a program and the overarching programmatic elements, such as commitment to diversity, the inclusion of those with lived experience, the flexible admissions policies and procedures, interdisciplinary faculty, flexible team, administrative structure, and valuing of technology in conducting research, learning, and teaching [28].

Curriculum co-designing and co-teaching: these are activities considered “experiential” and “collaborative” because they enhance the critical and reflexive engagement with the reading material and develop an appreciation for the difficulties involved in designing a syllabus and translating it into effective classroom time [29,30]. The process generates discussion between the faculty and the student co-designers and builds support and commitment among the students. Walters and Misra and other authors also assert that curriculum co-designing allows the faculty to pay close and individualized attention to each student project [29,30,31].

In UWW, some elements of the curriculum were set by the core team, and others were co-developed. The UWW co-directors chose the main focus of each fellowship cohort based on a) the needs emerging from the field, b) the requirements of the funders (based on priority research areas), and c) the specific research needs of the fellows (including how to present online and using online tools for knowledge transfer and exchange). Engaging with the fellows on their research needs allowed them to shape the curriculum and co-deliver it. For example, this co-designing (as well as making the implicit curriculum visible) is central in research with Indigenous (First Nation, Métis and Inuit) communities in Canada in order to make visible the colonial aspect of our research tools. In UWW, our (largely non-Indigenous) fellows received support, constructive criticism, and practical tools from each other and from invited researchers in that field to learn to approach and sustain their partnerships with Indigenous communities.

Thus, community-based research, interdisciplinarity, intervention research, and applied ethics function more like a set of values than boxes to be filled with specific content. At the start of each cohort, the curricular items of interest were gathered, a common calendar of activities was co-designed, and fellows were required to present, teach, and prepare summaries of research or research problems (theoretical, institutional, logistical, methodological). The result was a fresh version of the UWW curriculum each year while retaining its core elements. UWW fellows were encouraged to broaden their horizons to propose ideas and learning activities that were beneficial to their educational and career development but keep in mind the welfare of the communities we serve and the welfare of all the fellows.

The community of practice: Following Jean Lave and Etienne Wenger’s [32] widely accepted views on situated learning as “the mastery of knowledge and skill requires newcomers to move toward full participation in the sociocultural practices of a community” (p.29) encouraged us to sustain the community of practice online component of UWW. Our frequent online sessions (bi-weekly, from 2 to 3 h online), of which a number had community and academic guests, and our personalized invitations to fellows to join us in webinars and other online and in-person events, provided a safe space where emerging academic and community fellows could describe their work and their struggles to develop and sustain their research, issues around research funding, institutional and community politics, and the job-seeking process. As well, the UWW community of practice provided support for fellows to engage with local organizations and take risks. As mentors, we often encountered shyness and self-doubt among fellows to approach community leaders, established academics, and even meet new peers. The online setting provided a space where fellows could be candid about anxiety over first impressions. Online meetings also allowed us to enhance the matches we jumpstarted as field mentoring placements, often after a great deal of negotiation with, and coaching of, the site supervisors.

### 3.2. UWW—Key Themes from Focus Groups and Interviews

In the following section, we present some of the main themes related to UWWs’ functioning and principles as distilled from focus groups, interviews, and open-ended responses from the pre-, post, self-assessment surveys, supplemented with our observations of subsequent UWW cycles. 

#### 3.2.1. Blended Learning, Online Presence and Networking

A significant “soft outcome” of synchronous online learning was that online sessions became a space where fellows were able to develop a “social presence” as a precursor to lasting professional connections. The literature tells us that the investment and outcome of developing a social presence online might be higher than the actual cognitive presence [33]. As one fellow described:

“The technology works amazingly well, and it brings us all together across the country in a way that feels quite real. We are private chatting and follow up with each other later about stuff that comes up. It is the thread that keeps it all together at regular intervals.”

This sort of experiential learning tended to be overlooked but often had a lasting impact on UWW fellows in the long term. A fellow from an early cohort remarked: 

“What I really enjoyed about this experience was being able to network with my colleagues, get to know their areas of interest, and be a part of a larger network of health researchers, which includes both past, present, and future UWW fellows. That opportunity to network and converse with my colleagues was priceless.”

Fellows’ increased proficiency with online learning was an important soft outcome of UWW. One fellow stated that “the use of technology was new for me. It got me learning more about technologies”. Another fellow encouraged future generations to utilize technology “to be creative pedagogically, e.g., use of chat room features in some of the presentations allowed for the spontaneity and real-time vibe of ‘classroom’ learning”. As e-learning continues to rapidly develop and grow as a tool within the field of higher education, these soft outcomes related to fellows’ comfort online have translated into more measurable outcomes in terms of employability and developing a valuable skillset. Since we started in 2009, given the 2020 restrictions on in-person teaching and other learning events, we are seeing an exponential growth in the appreciation of the impacts of graduate online learning and mentoring.

#### 3.2.2. Learning to Experience Community-Based Research

People living with HIV in the 1980s were among the first committed HIV researchers in western countries due to the initial neglect from the North American social institutions that should have mobilized a response to HIV [34]. This legacy of HIV activism in research informs the emphasis that UWW placed on community-based research principles—a process that connects researchers and community members in order to conduct collaborative research that addresses a community issue or problem. The fundamental question underlying the development of CBR in the field of HIV is not about the content of knowledge but, rather, *what knowledge is for* [35]. Community-based HIV research is rooted in an ethical engagement with the community and permits a high level of engagement between disciplines, as the approach incorporates diverse perspectives, concepts, and theories and/or tools/techniques and/or information/data [36]. UWW’s approach to CBR took to heart the values of engaged scholarship.

The UWW program has always prioritized two main principles that underline CBR and engaged scholarship in HIV research in Canada: The Greater Involvement of Persons Living with HIV/AIDS (GIPA) principles, which promote the greater and meaningful inclusion of people living with HIV/AIDS [37], and the Canadian Tri-Council Policy Statement: Ethical Conduct for Research Involving Humans (TCPS2) principles, which outline strategies for centering respect for persons, concern for welfare, and justice when conducting research with indigenous (First Nation, Métis, and Inuit) peoples—the original stewards of the land [38]. Together, these two ethical frameworks foreground the rights of people being “researched” and support them in becoming full partners and beneficiaries of research done with, on, and for their communities. HIV researchers in Canada have sustained a strong practice of supporting people living with HIV working in all aspects of HIV research [39,40,41,42,43,44]. To maintain this direction, in each UWW cohort, we included a number of “community scholars” and seasoned people living with HIV as “peer researchers”. Often, community leaders and even clinical staff have a more limited access to travel and conferences. Technology allowed us to foster collaborations between AIDS leaders, frontline staff and emerging researchers; online learning can be an equalizing agent [45].

In the post-training assessment of the fifth UWW cohort, one fellow reported:

“The most significant change in knowledge that I have experienced relates to CBR as it relates to HIV in particular. I think I have a much deeper understanding of the particular needs of academic researchers, ‘community members’, and community researchers in collaborative projects. I also have a deeper appreciation of the stakes surrounding people from ‘community’ partaking in research.”

These reports were not uncommon among the fellows and reflected the general learning outcomes in this area over time.

#### 3.2.3. Experiencing Community, Intersectionality, and Interdisciplinarity

Kolb stated that experiential learning includes the process of cultivating relationships, engaging in co-learning, and engaging with—and perhaps becoming part of—various communities [46]. One of the primary ways that the UWW program consistently broke down walls between the graduate studies’ experience and the complexity of HIV community-based organizational context was through fellows’ field mentoring placements, a form of community service learning practicum with either a non-profit, policy, or academic environment [47]. Our UWW tweak: connecting the student with a field mentoring opportunity that was challenging for them and not an obvious fit for their chosen program of studies. For example, a student with predominantly a social theory background was placed with a needle exchange service in a different province, and we asked a social work student—inclined to pragmatic and political action—to work in the slower and complex world of policy-making. Such pairings were not always easily accepted and required negotiations with fellows and faculty over the phone, email, and web conferencing check-ins. But they did provide valuable learning and networking opportunities for fellows.

It is not coincidental that over twenty of the 63 UWW fellows, from 2009 until 2015, have self-identified as LGBTQ2SI* (i.e., lesbian, gay, bisexual, transgender, queer, two-spirited, intersex, and allies), and at least six fellows identify as living with HIV. Thus, UWW offered a safe space to experience these identities and prepared for future audiences, organizations, and projects. In general, the fellows selected to participate in UWW were open about their identity (including being HIV-positive), their political values, their street-level activism, and their struggle to reconcile these with institutional and professional values. Indeed, exploring personal and political identities and roles has been one of the living, implicit components of the UWW curriculum. This particularly happens when the learning activities are co-produced by established scholars and graduate students [48].

In addition to fostering lasting research relationships between researchers and community members, UWW also concentrated on connecting fellows across disciplinary boundaries. Fellows were encouraged to work on research teams that employed a variety of disciplinary perspectives and research or evaluation methods [49]. Thus, one of UWW’s most important contributions was its ability to use technology as a “liminal learning space” [50] between the concrete every day of the student, the concrete every day of community organizations, and the broader disciplinarian thought, so learners could interact and challenge each other to think about how to address problems and move to feasible solutions. The program emphasized the importance of fostering a community of learners who understand different disciplines and perspectives. Most UWW fellows entered the program with backgrounds in three to five academic disciplines [51]. The wide range of academic perspectives within UWW helped fellows become more familiar with the processes of interdisciplinary research. As one fellow described it, “the opportunity to dialogue with the other fellows, our invited guests, and others connected to the vast UWW network allowed me to gain both a new appreciation for and understanding of approaches relevant to uses of interdisciplinary research.”

The interdisciplinary setting that UWW cultivated helped fellows to situate themselves and their work in other interdisciplinary environments as well. As one fellow explained, “the [field mentoring] placement helped me to see how, with my skills and training, I can be an integral part of an interdisciplinary research project. It clarified for me the value of my experience and strengthened my confidence in my community-based research skills”.

### 3.3. UWW Fellows’ Self-Assessments

UWW fellows were asked to assess their knowledge and skill levels throughout their fellowship and to track the amount of knowledge transfer and exchange products that they produced [52]. These self-assessed changes demonstrated the contributions that the training made to fellows’ awareness, knowledge, and practice (of methods, analysis tools, etc.). They also highlighted the “softer” outcomes of the UWW training, such as the influential learning experiences that fellows were exposed to and the types of communities that fellows were able to connect with.

In this final section, through a presentation of both the self-assessment change scores and the UWW fellows’ open-ended responses to the self-assessment questions, we illustrate that these “soft outcomes” are integral pieces of the development of up and coming HIV CBR researchers. It has been emphasized that while the “soft outcomes” of training programs, such as interpersonal, organizational, analytical, and personal skills, might seem insignificant, the leap forward in achieving these outcomes could be immense and might mark progress towards employability or other more measurable outcomes [53].

The variety of *types* of experiences that fellows had throughout the program was an indicator of UWW’s effectiveness. Fellows were asked to rate their progress in levels of knowledge (content), skills (application) and experience using a scale from “novice/advanced beginner” to “proficient/expert” before and after their fellowship developed by Brocklehurst & Rowe [54]. The most dramatic changes in fellows’ level of experience occurred within three areas. First, fellows reported a large increase in their level of experience working with a cross-section of persons living with HIV. As one fellow reported in their self-evaluation, “being part of UWW has been a great way to network and engage with researchers, community members, and service providers from populations I didn’t know much about”. Second, fellows reported significant increases in their level of experience working in the non-profit sector. Fellows emphasized that their placements within community organizations were important opportunities to expand their range of experience in HIV research. One fellow reported, “the fellowship provided the opportunity to meet, engage with, and build relationships with a broad array of HIV-related organizations and people living with HIV that was much broader than my past (narrow/more specific) experiences”. Another fellow explained how connecting with non-profit organizations would have a long-term impact on professional development: “The fellowship gave me the opportunity to collaborate with a new set of practitioners and non-profit organizations, some of which have agreed to be a partner in my Ph.D. research and beyond”. Third, fellows reported a large increase in their level of experience within Canadian Indigenous settings. One fellow was confident that “I significantly improved my knowledge about CBR and Aboriginal HIV research”. Another fellow acknowledged that the opportunity to gain experience in Canadian Indigenous settings helped him to gain confidence working in the area: “I think that by the end of UWW, I was feeling more comfortable being a better ally then I was before”.

## 4. Limitations

As a reflective case study of a training program that we were an integral part of, the co-authors brought our preconceptions and positive biases to UWW’s assessment. However, fellows’ self-evaluation (and annual program evaluation) data were collected by an independent assessor and anonymized to the extent this was possible. The quantitative self-evaluation data were not included in this report, but supported our claims about this cohort-learning process. We relied on the fellows’ voices and perspectives as much as possible in this case study to minimize painting too positive a picture of UWW processes and outcomes. Indeed, fellows were vocal about some of the elements of UWW that were not as successful; for example, in the early years of UWW, online webinar technology was unreliable, and fellows expressed fairly vocal dissatisfaction with this element of the program. (However, since the quality and ease-to-use of web-conferencing and educational online tools has increased rapidly in the last decade, we did not account for this positive and generalized change here).

In terms of transferability, for its first ten years of functioning, UWW had dedicated, external funding and was designed as a training program to function outside of a traditional academic environment (hence the name “Universities Without Walls”). Thus, UWW did not have constraints of program or curriculum approvals, and was able to adapt quickly and experiment with training components and provide a collaborative, flexible learning environment with the fellows. While translating elements of the UWW program to a post-secondary environment would certainly be possible, it is unlikely that the UWW program could be transferred to such an environment in its entirety. UWW was also built for the socio-political environment of HIV health research, and, thus, some of the components of UWW would not necessarily be appropriate for training endeavors in other areas.

## 5. Conclusions

As research and activism dedicated to HIV/AIDS continue to adapt to the changing socio-political contexts of the virus, the UWW program demonstrates that blended training with a strong dose of mentorship and technological support can produce researchers well-connected to their chosen career path and their communities, community-grounded, and yet aware of the demands of academia. Online learning (synchronous and asynchronous) should not be compared with classroom teaching—the comparison has been demonstrated not to hold— however, it offers similar learning opportunities while demanding sustained focus, preparation, and additional skills development [54].

The experiences of those involved with UWW demonstrate that explicit curricular components—such as interdisciplinarity, community-based research, intervention research, and applied ethics—can be learned through a blended delivery when combined with opportunities to apply the knowledge in ways, such as a field mentoring placement and a learning institute. Related to this outcome, our case study described that implicit curricular components in the formation of a professional—the sense of self in the field as a researcher, student, and community member—can also be delivered by a blended model. However, the tools and activities need to be tailored to each student for their context while pushing their disciplinarian and professional boundaries. Most significant is the case of having built a liminal learning space in-between a) rigorous academic contents and practices (e.g., following ethics, practicing interdisciplinarity) and b) intuitive and grassroots community processes (e.g., Indigenous elders’ and knowledge keepers’ knowledge, minority groups’ ethical practices). Our fellows have consistently understood that they will professionally, and even personally, live in these thresholds. In 2020, the three directors of UWW are reassured in these accomplishments when we see twenty UWW alumni hired in permanent faculty positions in Canada and nearly all of them working in the field of HIV research, clinical service or nonprofit.

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
