# Peer review of "Universities without Walls: A Blended Delivery Approach to Training the Next Generation of HIV Researchers in Canada"

_ijerph, 2020, doi:10.3390/ijerph17124265_

Round 1

Reviewer 1 Report

This is an important and significant contribution not only to HIV/AIDS education, but also to online pedagogies and the strengths of good, meaningful online cohort-based learning. A particular strength of the article is the presentation of what is a rich and varied data set, as well as the analysis that was carried out. In a time when universities have had to shift rapidly to online education, I would think that an article such as this is would be a valuable resource, especially an object lesson for innovative, blended and online pedagogies. That the paper also addresses directly preparing researchers to work with and in Indigenous communities is another strength and the comment from one of the participants a power endorsement of the project (lines 409-410).

In terms of readability - it is well written and integrates theory and related literature quite well. One small change that is necessary is to put quotes around participant feedback. That should happen on lines 279-287 (and was that just one person or were those multiple comments from multiple people? and on lines 224-227. 

Finally, I very much appreciated that the article's claims did not exceed the evidence provided. I think this is ready for publication, and I look forward to being able to direct students and colleagues to the paper. 

Author Response

Point 1

Reviewer: One small change that is necessary is to put quotes around participant feedback. That should happen on lines 279-287 (and was that just one person
or were those multiple comments from multiple people? and on
lines 224-227.

Response: thanks for noticing those important omissions, those changes have been made in the lines indicated.

Many thanks

Francisco

Reviewer 2 Report

Dear Authors,

Thank you for your submission and the well written manuscript. I have become convinced that the paper has made the case for the so-called blended training approach.

I have only two minor questions, which I believe the authors will realize their importance and will address them during their revision:

  1. There must be some limitations to this discussion. It seems that you have omitted this.
  2. In this qualitative study, toward the end the broader audiences will appreciate your further discussion on the possibility of building an empirical strategy so that the knowledge can become applicable and bear policy implications, regarding the training approach.

Also, please spend some time on the "dark side" of the real-world practices, as it is an integral part of the industry, e.g., this one reflection: https://www.mdpi.com/2077-0383/7/11/402

Generally speaking, I believe this can be a valuable contribution and hope that the authors will spend time perfecting their paper, and that the above comments might be helping.

I am rendering my recommendation: Minor revision.

Author Response

Dear Reviewer 2

Point 1

  1. There must be some limitations to this discussion. It seems that you have omitted this.

Many than ks for this feedback. Please note the addition of a "limitations" section from lines 410 to 433. They address limitations of qualitative data and potential transferability of a cohort-driven program for emerging researchers.

Point 2

  1. In this qualitative study, toward the end the broader audiences will appreciate your further discussion on the possibility of building an empirical strategy so that the knowledge can become applicable and bear policy implications, regarding the training approach. Also, please spend some time on the "dark side" of the real-world practices, as it is an integral part of the industry, e.g., this one reflection: https://www.mdpi.com/2077-0383/7/11/402

Response: as suggested, in the "conclusions" section, we address the challenge of building a liminal learning space where students learn by the rules while observing the localized ethics and rules of community groups, ethno-racialized groups and sexual minorities/minorities with intersecting identitites.

Many thanks for the thorough read,

Francisco 

Reviewer 3 Report

I have no comment

Author Response

Dear Reviewer 3: many thanks for taking the time to review our manuscript

Francisco